# Environment DNA Reveals Fish Diversity in a Canyon River within the Upper Pearl River Drainage

**DOI:** 10.3390/ani14162433

**Published:** 2024-08-22

**Authors:** Si Luo, Meng Wang, Weizhong Ma, Dangen Gu, Zhijun Jin, Ruiqi Yang, Zhen Qian, Chengwen Song, Zexin Wang, Shiyu Jin

**Affiliations:** 1School of Life Science and Food Engineering, Huaiyin Institute of Technology, Huai’an 223003, China; luosi@hyit.edu.cn (S.L.); 13258004185@163.com (R.Y.); 15348648151@163.com (Z.Q.); songchengwen2008@126.com (C.S.); 18303615694@163.com (Z.W.); 2Power China Guiyang Engineering Corporation Limited, Guiyang 550081, China; perfectwm2022@126.com (M.W.); weizhongma123@sina.com (W.M.); jinzj_gyy@powerchina.cn (Z.J.); 3Pearl River Fisheries Research Institute, Chinese Academy of Fishery Sciences, Guangzhou 510380, China; gudangen@prfri.ac.cn

**Keywords:** canyon rivers, eDNA, fish diversity, dams, dominant species

## Abstract

**Simple Summary:**

The construction of dams in Mabiehe River has significantly impacted fish diversity, resulting in reduced species richness and diversity within dammed sections compared to undammed segments of the river. The environmental DNA (eDNA) technique is effective in assessing fish diversity in the challenging environment of canyon rivers. The successful detection of a variety of species, including both protected and exotic ones, highlights the robustness of eDNA as a comprehensive monitoring tool.

**Abstract:**

Investigating fish diversity in canyon rivers through conventional fish surveys is challenging due to precipitous conditions, including steep slopes, rapid water flow, and complex habitats. Additionally, intensive construction of dams has further complicated the understanding of contemporary fish diversity in these rivers. In this study, we used the environmental DNA (eDNA) technique to assess fish diversity and examine the effects of dams on fish diversity in the Mabiehe River, a canyon river in the upper reaches of the Pearl River drainage. Water samples from 15 sampling sites were collected, yielding 9,356,148 valid sequences. Utilizing the NCBI public database, a total of 60 freshwater fish species were identified, with *Carassius auratus*, *Cyprinus carpio*, and *Pelteobagrus fulvidraco* being the most dominant species in the Mabiehe River. We also detected one nationally protected fish species, three provincially protected fish species, and six exotic species in this river. Furthermore, eDNA analyses demonstrated that the lotic river sections harbor more species and greater diversity than dammed sections, suggesting that dams might exert significant impacts on local fish diversity. Overall, this study supports the effectiveness of the eDNA technique as a complementary tool to traditional field surveys for monitoring fish biodiversity in canyon rivers.

## 1. Introduction

Canyon rivers often have unique features. These include steep gradients, fast-flowing currents, rich hydraulic resources, and complex habitats. These characteristics make canyon rivers vital for both water resource management and hydropower development, as well as essential habitats for native fish species [1,2,3,4,5]. The construction of high-strength dams in canyon rivers is widely recognized to significantly affect local fish diversity [6]. However, the empirical evidence for these impacts remains limited within these river systems. There are insufficient data on the long-term impacts of dams on fish diversity. Studies provide snapshots of biodiversity at single points in time, without examining how fish communities change before and after dam construction, or how they might fluctuate over extended periods. The formidable conditions in canyon rivers make conventional fish survey methods challenging. For instance, the steep and rugged terrain can make access to certain areas difficult, while the fast-flowing currents pose a risk to both equipments and researchers. Additionally, the complex habitats with various submerged structures can impede the use of traditional netting and trapping techniques, often leading to incomplete or biased sampling. These conditions necessitate labor-intensive and invasive sampling methods that may not adequately capture the full diversity of fish species present.

To address these challenges, it is imperative to employ appropriate methodologies that can accurately assess fish diversity and the influence of dams on local fish populations in canyon rivers. The environmental DNA (eDNA) technique offers a promising alternative by effectively detecting the presence of the target organisms through the analysis of DNA released into the water. This noninvasive and sensitive method provides a significant advantage over conventional methods, particularly in difficult-to-access environments like canyon rivers [7,8,9].

The Mabiehe River, situated in the upper reaches of the Pearl River in Guizhou Province, China (Figure 1), represents a valuable opportunity for investigating the impacts of dams on local fish diversity in canyon river ecosystems. Presently, the Mabiehe River hosts ten dams, significantly modifying fish habitats [10], which appears to be a substantial determinant affecting fish diversity in this river [11,12,13,14,15]. Despite these challenges, only one relevant study has been conducted in this river to date [16]. That study reported that the upstream area of the Maling dam exhibited comparatively higher species richness and greater diversity indices than the downstream area, as determined through conventional fish surveys, suggesting that the Maling dam has affected fish diversity to some extent. However, as it solely focused on a single dam and its immediate adjacent river segments, comprehensive knowledge regarding fish composition, diversity, and the broader impacts of dams on fish diversity within this river system remains limited.

The eDNA technique has proven valuable in elucidating the distribution of aquatic organisms and assessing biodiversity, particularly in studies focusing on fish communities [17,18,19,20,21,22,23]. By analyzing eDNA, researchers can obtain a more comprehensive and accurate picture of fish diversity and distribution, even in challenging environments like canyon rivers. This study employs the eDNA technique to conduct an initial assessment of fish composition and diversity across 15 segments of the Mabiehe River. These sampling locations encompasses both upstream and downstream areas of dams, as well as sections of the lotic river unaffected by dams. Moreover, the current study aims to elucidate the potential effects of dams on indigenous fish diversity. This study represents a significant endeavor in utilizing the eDNA methodology to monitor fish biodiversity in a canyon river setting.

## 2. Materials and Methods

### 2.1. Study Region, eDNA Sampling, and Sample Preservation

The study region is the Mabiehe river, situated within the upper reaches of the Pearl River basin and characterized by a north-to-south flow over a length of approximately 140 km. A total of 15 sampling sites were established within this river segment (25.049° N to 25.556° N, 104.847° E to 104.991° E; Appendix A; Figure 1), comprising seven sites in the upstream area of dams (USD), four sites in the downstream area of dams (DSD), and four sites in the lotic river affected by dams (LTR) (Figure 1).

At each sampling site, 2 L of water samples were collected from three different depths (the surface, 0.5 m, and the lower layer) using a water collector positioned approximately 0.5 m below the water surface in August 2022. These samples were then mixed to create a composite sample that more accurately represents the vertical biodiversity profile of the river. Three replications were conducted at each sampling site. Prior to sampling, water samplers and sampling bottles were disinfected with 10% bleach powder solution, and sampling personnels changed disposable gloves after each collection. To prevent DNA degradation, the water samples were immediately stored in cold storage and filtered using a vacuum pump onto a 0.45 μm mixed cellulose filter membrane (Whatman, Buckinghamshire, UK) within 12 h. The vacuum pump (JOANLAB, Huzhou, China) was sterilized and rinsed before each filtration to avoid cross-contamination. Subsequently, the filter membranes were promptly transferred to sterile 4.5 mL plastic scintillation tubes and rapidly stored in liquid nitrogen in the field. Finally, the filter membranes were preserved at −80 °C in the laboratory until DNA extraction.

### 2.2. DNA Extraction, Amplification, and Sequencing

The filter membranes were used to extract the total DNA of the water samples utilizing the PowerWater DNA Isolation Kit (Qiagen, Hilden, Germany) following the manufacturer’s instructions. Subsequently, the quality of the extracted eDNA was examined using 1% gel electrophoresis. Furthermore, each sample was extracted independently and a negative control with a blank filter membrane was set. The extracted DNA was then preserved at −80 °C in an ultra-cold storage freezer (Haier, Qingdao, China).

Given that past studies have demonstrated that primers targeting the *12S rRNA* gene generally outperforms other primers targeting the other genes, such as *16S rRNA* or the mitochondrial cytochrome c oxidase I (*COI*) gene, in terms of amplified fish diversity [24,25], we thus choose the fish universal primers Tele02-F (5′-AAACTCGTGCCAGCCACC-3′) and Tele02-R (5′-GGGTATCTAATCCCAGTTTG-3′) targeting the 12S rRNA region of the mitochondrial genome for PCR amplification in this study [26]. This primer is an improved version derived from the “MiFish-U” primer and has been demonstrated to enhance taxa resolution [26]. This process uses a 20 μL reaction system comprising 4 μL of 5 × FastPfu Buffer (Transgene Biotech, Beijing, China), 2 μL of deoxynucleoside triphosphates (Transgene Biotech, Beijing, China), 0.4 μL of FastPfu Polymerase (Transgene Biotech, Beijing, China), 2–5 μL of template DNA, and 0.8 μL of forward and reverse primers. The PCR reaction conditions included the following five steps: pre-denaturation at 95 °C for 5 min, 35 cycles of 95 °C for 30 s, 58 °C for 30 s, and 72 °C for 45 s, and a final extension step at 72 °C for 10 min. To test whether contamination occurred during the PCR amplification, the PCR negative control was set using a ddH2O template. The PCR products were checked by 2% agar gel electrophoresis. Subsequently, the purified PCR products were recovered and used to construct the Illumina PE250 library. Finally, sequencing was performed by the Lingen Biotechnology Co., Ltd in Shanghai, China, using paired-end sequencing on an Illumina NovaSeq 6000 System (Illumina Inc., San Diego, CA, USA).

### 2.3. Bioinformatics Processing

Poor-quality sequences with a length shorter than 100 bp were eliminated using Trimmomatic v.0.36 [27], and then, pairs of reads were combined into a sequence using FLASH v.1.2.11 [28]. The chimeric sequences were deleted, and high-quality sequences were combined as “parent–child” sets with 97% similarity using Usearch version 10 (http://drive5.com/uparse/, accessed on 1 August 2024) [9]. Subsequently, the yielded unique sequences were used for comparison and taxonomic annotation on the NCBI database (https://www.ncbi.nlm.nih.gov/, accessed on 18 July 2024) using the BLASTn tool. A preliminary taxonomic annotation table was obtained by searching the unique sequences on the NCBI database according to the following criteria: ≥97% similarity, e ≤ 10^−5^, and coverage ≥ 0.9 [29]. Molecular operational taxonomic units (MOTUs) targeting the same species were merged. MOTU sequences corresponding to non-fish species and fish species not expected to inhabit the region were manually removed based on historical survey data from the upper Pearl River including the monograph “Ichthyography of the Pearl River, Ichthyography of Guizhou, China, and Species diversity and distribution of inland fishes in China”.

### 2.4. Statistical Analyses

To calculate diversity indexes, we randomly selected reads from each sample to be normalized based on the minimum number of sequences yielded in any of the samples utilizing QIIME v.1.9.0 [30]. This normalization procedure maintained a consistent relative sequence abundance for each species in each sample. Firstly, the fish species composition was depicted through the generation of bar charts by virtue of the detected relative sequence abundances of species. Subsequently, two diversity indexes including alpha diversity and beta diversity were calculated. With respect to alpha diversity, we chose the Shannon index, Gini–Simpson Index, and Pielou index. The Shannon index reflects the diversity, and the Simpson and Pielou indices synthetically represent evenness of the species [8,17]. For beta diversity analysis, the principal co-ordinates analysis (PCoA) and non-metric multidimensional scaling (NMDS) analysis were conducted based on the Bray–Curtis distance.

## 3. Results

### 3.1. Fish Composition

A total of 9,356,148 valid sequences were obtained from 15 sampling locations. After annotation and screening, the samples collectively revealed 60 fish species, which belonged to 10 orders, 20 families, and 52 genera (Table 1). Among the determined species, Cyprinidae (*n* = 35) demonstrated the highest species abundance, accounting for 58.33%. The remaining families comprised no more than three species each (Table 1). Notably, our study interestingly found one nationally protected fish species (*Percocypris pingi pingi*) and three provincial protected fish species (*Pareuchiloglanis longicauda*, *Acrossocheilus longipinnis*, and *Semilabeo obscurus*) in Guizhou Province (Table 1). Furthermore, six alien species (*Acipenser baerii*, *Ictalurus punctatus*, *Clarias gariepinus*, *Coptodon zillii*, *Lepomis cyanellus*, and *Gambusia affinis*) were also identified (Table 1). Remarkably, three alien species, namely *A. baerii*, *C. zillii*, and *Gambusia affinis*, were detected in over 70% of the sampling sites (Figure 2). An analysis of the relative sequence abundance revealed *Carassius auratus*, *Cyprinus carpio*, and *Pelteobagrus fulvidraco* as the top three species, evenly distributed across all sampling sites (Figure 3). Additionally, the major carp *Ctenopharyngodon idella* was detected in overall water samples with relatively higher sequence abundance (Figure 3). In total, 23 species were identified across all sampling sites (Figure 2).

With respect to upstream (USD) and downstream (DSD) dams, the results showed that the number of species of the upstream dams and downstream dams ranged from 32 to 37 and from 38 to 47, respectively (Table 2). In the lotic river section without dams (LTR), the count of species ranged from 38 to 44 (Table 2). It was observed that sampling sites DSD and LTR hosted a greater number of species compared to upstream dams (Figure 4a). Additionally, a comparable fish composition was observed, with 75% of species shared among the three groups (Appendix A).

### 3.2. Alpha and Beta Diversity Analyses

The coverage of all sampling sites exceeded 99.0%, indicating that the sequences accurately reflected the true diversity of the water samples. The Shannon index values ranged from 2.47 to 2.85 across 15 sampling sites (Table 2). In addition, the Gini–Simpson and Pielou indices for each sampling site ranged from 0.86 to 0.91 and from 0.69 to 0.77, respectively. The results indicated that LTR exhibited the highest values in the Shannon, Gini–Simpson, and Pielou indices (Figure 4b–d).

The NMDS analysis yielded a stress score of 0.1923, suggesting that the results had some explanatory significance. PCoA revealed that four sites within the LTR group could be distinguished from remaining sites and that BX and XTB within the DSD group were scattered and far apart from the rest of the sampling sites (Figure 5a). The NMDS results showed that the CLHD site in the LTR group and LYZ in the USD group were scattered and far apart from the other sampling sites (Figure 5b).

## 4. Discussion

### 4.1. Fish Composition and Community

The Mabiehe River is a highland and precipitous river where traditional fish investigation techniques are challenging to implement. Consequently, fish surveys in this river have been scarce, and data on fish community are limited. In our study, we identified a total of 60 freshwater fish species using the eDNA approach, which outdistanced the species number observed in field surveys conducted with the traditional technique during 2018 and 2019 (22 species [16]) and during the 1980s (42 species [31]). Numerous studies have demonstrated that the performance of eDNA in fish detection can be superior to conventional methods [20,32,33]. In addition, Cyprinidae species were the dominant taxa in the Mabiehe River, constituting 58.33% of the identified species. This outcome was consistent with the findings from the field surveys in the 1980s [31] and recent field survey during 2018 and 2019 [16]. These results demonstrate that the eDNA approach is an effective method for revealing fish diversity in this canyon river.

Our observation indicates that *Carassius auratus*, *C. carpio*, and *P. fulvidraco* are likely the most dominant species in the Mabiehe River. These species can be easily captured in the Mabiehe River and/or its adjacent waterways using traditional methods [16,34], suggesting that they possesses relatively large biomass. Notably, a major carp *C. idella* was found in all water samples with relatively high sequence abundance. This finding is likely attributed to artificial release activities, as the river does not provide the necessary conditions for the completion of its life cycle. Multiple artificial releasing activities containing this species have been conducted every year.

Among the 60 identified species, we unexpectedly detected 1 nationally protected fish species (*P. pingi pingi*) and 3 provincially protected fish species (*P. longicauda*, *A. longipinnis*, and *S. obscurus*), which were rarely captured in conventional field surveys. *P. pingi pingi*, *P. longicauda*, and *S. obscurus* were only found at one, two, and three sampling sites, respectively. Furthermore, the sequence abundances of *A. longipinnis* at the sampling sites where this species was detected were extremely low. Compared with the study (traditional fishing methods) by [16], which covered the region from CLHD to MLZ (as illustrated in Figure 1 of our manuscript), our eDNA analysis detected a significantly higher number of species. Specifically, the traditional fishing methods used by [16] failed to detect 42 fish species that were identified using eDNA in our study. Only the following species were detected in both studies: *Abbottina rivularis, Pseudorasbora parva, Hemibarbus labeo, Hemiculter leucisculus, C*. *carpio, Carassius auratus, A*. *yunnanensis, Onychostoma gerlachi, P*. *pingi, Discogobio tetrabarbatus, S*. *obscurus, Schizothorax lissolabiatus, Schistura fasciolata, Misgurnus anguillicaudatus, Paramisgurnus dabryanus, Silurus asotus, P*. *longicauda*, and *Rhinogobius similis*. This significant discrepancy highlights the effectiveness of eDNA in monitoring fish biodiversity, especially in challenging environments like canyon rivers where traditional methods may fall short. However, it was worth noting that our eDNA analysis did not detect four fish species (*Rectoris longifinus*, *Sinocrosscheilus tridentis*, *Sinocyclocheilus angustiporus*, and *Pterocryptis anomala*) that were identified by [16]. These species typically have smaller biomass and may be underrepresented in eDNA samples due to their specific habitat preferences and lower overall abundance. Numerous studies have demonstrated that eDNA can be used to monitor endangered species and provide insights into the status of endangered species, which are often difficult to capture using traditional methods [22,35,36,37]. Our study further supports this conclusion.

In the current study, we unexpectedly detected six exotic species, which accounted for 10% of the total observed species. With respect to *A. baerii* and *I. punctatus*, the presence of these species in water samples was likely due to the aquacultural effluent entering the river, as they were not detected in recent field surveys. These species are economically valuable and widely cultured in various farms [38]. The flow of aquacultural water into the Mabiehe River led to the detection of these species through eDNA analysis in our water samples. It is also worth noting that *A. baerii* is intensively farmed in the Guizhou Province, which may explain its presence in 11 out of 15 water samples. The other economic species, *C. gariepinus*, was detected only at the most downstream sampling site (ZJD), likely due to the artificial culture in the Wanfeng Lake Reservoir. This reservoir, which is close to our sampling site, supports large numbers of *C. gariepinus*, making its detection in the ZJD water sample expected. *Lepomis cyanellus*, previously reported to invade China [39,40], was detected in our recent field surveys in Guizhou Province (unpublished data). Although the sequence abundances of *L. cyanellus* in six sampling sites were relatively low, local governments still raised concern about its potential for large-scale invasion. We also found that *C. zillii* and *G. affinis* were prevalent across all water samples, which was indicative that these species may pose the most significant threat as exotic species in the Mabiehe River. These two species were also found in this river during our recent field investigations (unpublished data). Given that *C. zillii* and *G. affinis* are two widespread invasive species in China [41,42,43,44,45,46,47,48], they can outcompete native fish for resources, alter habitat structures, and disrupt existing food webs. This can lead to a decline in native species populations and a loss of biodiversity. These changes can compromise ecosystem services such as water purification, nutrient cycling, and recreational fisheries, ultimately affecting the ecological integrity and sustainability of the river ecosystem. Increased efforts in developing targeted management strategies to protect native biodiversity and maintain ecosystem functions should be made to minimize their invasion areas as much as possible.

### 4.2. Effects of Dams on Fish Diversity

This study found that the CLHD site had the highest value of the Shannon index and the second largest values of the Simpson and Pielou indices, indicating higher fish community diversity at this site. In addition, the NMDS analysis showed that the CLHD site was distinct and far apart from the other sampling sites, implying low similarity in fish composition compared to the other sites. The CLHD site is characterized by a lotic environment with minimal artificial interference, and its surrounding regions contain several karst underground rivers. These diverse environmental forms offer multiple habitats for harboring more fish species and a stable community structure. Moreover, considering that this site has experienced minimal artificial disturbance (e.g., dams), the fish composition and community structure largely remain in their original condition, which leads to significant differences in fish composition between this site and the other sites.

The most salient finding in our study was that the sites in the lotic river had the highest species count and the largest alpha-diversity indices, suggesting that the lotic rivers might harbor greater fish richness and a more stable community structure than river sections with dams. Different fish compositions in the lotic rivers were also confirmed by PCoA. The observed higher diversity and evenness in the LTR section can be attributed to its status as a flowing river section that is less affected by dam-related disturbances. The original ecological conditions in the habitats are largely maintained, allowing for a more stable and diverse community of species. In contrast, sections of the river influenced by damming often experience altered flow regimes, habitat fragmentation, and changes in water quality, all of which can reduce species diversity and disrupt community composition. The distinct community compositions observed in the PCoA and NMDS analyses further reflect these environmental gradients and the varying degrees of anthropogenic impact across different sections of the river. Dams disrupt the continuous environment of river and directly affect the migration and dispersion of fish [49,50]. Even worse, some migratory fish are unable to complete their entire life cycles, which leads to a reduction or even disappearance of resources in dammed sections. In addition, the rising water levels in the upstream areas of dams and the slowing of water velocity changes the original homeostasis of the river and compress suitable habitats for fish, leading to declines in fish diversity and the number of fish species [11,51,52]. We also observed that downstream dam sites might have more fish species and richness than the upstream dam sites, which was likely due to the water samples collected at downstream dams including waters flowing from the upstream dams. These observed results indicate that dams in the Mabiehe River have remarkably influenced the fish diversity. Dams are obstructing rivers worldwide, impairing habitat and migration opportunities for many freshwater fish species. They remove turbulent river sections and create tranquil water bodies, thus affecting flow and temperature regimes, sediment transport, and species communities. The shift from lotic to lentic environments after dam construction often favors generalist species over specialists, altering the assemblages of taxonomic groups and putting endemic species at particular risk of extinction, leading to biotic homogenization [53,54]. Shifts in temperature regimes, including downstream decreases in temperature due to hypolimnetic releases from reservoirs, impair conditions favorable to native species but may favor exotics or habitat generalists [55].

The long-term ecological consequences of reduced fish diversity in dammed sections are significant. These changes can lead to impaired ecosystem services, such as nutrient cycling, water purification, and sediment transport, essential for maintaining ecosystem health. Furthermore, the reduction in biodiversity can negatively impact local fisheries, which rely on a diverse and stable fish population for sustainable yields. This can lead to economic losses for communities dependent on fishing for their livelihoods and diminish food security. Understanding these long-term consequences is crucial for developing effective conservation strategies and mitigating the adverse impacts of dam construction on river ecosystems [56].

### 4.3. Performance and Limitations of eDNA Technique in a Canyon River

In the present study, eDNA techniques reveals the fish composition, including endangered and exotic species, and evaluated the fish diversity in the Mabiehe River. Furthermore, eDNA analyses provided evidence that the lotic river sections harbored more species and greater diversity than river sections with dams. These outcomes underpin that the eDNA technique is an effective approach for monitoring fish biodiversity in a canyon river and complements traditional field surveys.

eDNA offers an expansive overview of biodiversity, but it has limitations in detecting all species present in an ecosystem. In the present study, two widespread species, *S. obscurus* and *S. tridentis*, and one endangered species, *S. angustiporus*, monitored in the Mabiehe River using traditional methods were not detected using the eDNA technique. These limitations arise from factors such as DNA degradation, primer specificity, and the dilution effect in large water bodies. DNA released into the environment by organisms disperses over space and degrades over time [57,58]. The persistence of eDNA is influenced by the density, life cycle features, species interactions, and size of the target species [59,60]. Biotic factors such as concentrations of bacteria and fungi, and abiotic factors such as nuclease activity, pH, oxygen content, conductivity, temperature, salinity, and ultraviolet exposure also affect eDNA persistence [61,62,63]. Fragment size is also crucial in the persistence of eDNA: fragments with 300–400 base pairs can persist for at least a week under controlled conditions [64,65], while shorter fragments (~100 base pairs) may last for months or even years [66]. Degradation limits eDNA studies, especially in warm, humid conditions, and affects inferences of fine-scale spatiotemporal trends in species and communities.

eDNA is characterized by highly heterogeneous DNA from various taxa or haplotypes, making full complementarity between primers and target sequences during PCR challenging [67,68,69]. This can lead to primer–template mismatches, affecting the stability of the primer–template duplex and the efficiency of polymerase extension, potentially causing biased results or complete PCR failure [69]. Primer bias can lead to the preferential amplification of abundant sequences over rare ones, shorter fragments over longer ones, or non-target organisms over target organisms [68]. Unlike metabarcoding, primer bias is not a major issue for barcoding, but it remains significant for eDNA detection. The targeted PCR-based amplification of samples using multiple sets of species-specific primers, instead of universal primers, should be strongly encouraged in eDNA barcoding, improving the accuracy of species detection [70,71,72]. The successful isolation and amplification of eDNA require the use of advanced sequencing technologies. Technologies like Illumina NovaSeq can detect more metazoan families compared to MiSeq, providing a more comprehensive view of biodiversity [73]. Selecting appropriate molecular markers for species identification, such as the mitochondrial cytochrome *b* gene [24,74], *COI* gene [75,76], and D-loop region [77], is also essential. In addition, although the “Tele02” primer has been demonstrated to enhance taxa resolution and has been widely used in eDNA studies focused on fish diversity [24,26,78,79], certain studies have indicated that this technique will still fail to detect particular species that can be observed using traditional methods [17,19,78,79]. For example, the eDNA method using the “Tele02” primer in our study did not monitor *Semilabeo obscurus* and *S. tridentis*, whereas traditional surveys can easily detect these species in many river sections [31]. Moreover, the limited length of amplicons can readily lead to species-matching failures and false positive results, particular for closely related species. Several other studies also used long (500–650 bp) barcodes to successfully detect species from eDNA [80,81]. Long barcodes may offer greater taxonomic discriminatory power, but long barcodes may suffer from reduced template concentrations. As a result, to address the limitations of the eDNA technique, it is imperative to employ multiple primers to examine both broad and specific taxonomic groups.

The choice of eDNA sampling and processing protocols significantly influences DNA yield, detection probability, and the resulting estimates of abundance and biodiversity [80,82,83,84]. In this study, we only collected water samples from inshore river sections and surface layers less than 1 m deep, making it difficult to detect underground river species *S. angustiporus* and benthic species *S. tridentis*. Different protocols were employed to collect the water, capture the eDNA with filters, transport the samples from the field, and store the water and/or filters prior to DNA extraction and amplification [85,86]. Immediate filtration of the water samples on site was a critical step to minimize DNA degradation. Preservation media such as ice, sodium acetate, lysis buffers, and absolute ethanol were used to stabilize the eDNA for enough time (up to 24 h) to safely transport it for storage and processing [86,87,88]. This reduces the exposure of the eDNA to environmental factors that contribute to its degradation, such as UV radiation and microbial activity [83,89]. Transporting samples in cold conditions and processing them promptly upon arrival at the laboratory further reduces the risk of eDNA degradation. This approach ensures the preservation of eDNA integrity from the field to the laboratory, enhancing the reliability of subsequent analyses. Field drying of the filters is an effective method for preserving eDNA, especially in regions where maintaining cold storage is challenging. Filters can be dried in the field before being sent to a lab, simplifying transportation and storage since no special precautions are needed, unlike for liquid handling [90,91]. This method has shown promising results in maintaining eDNA integrity, making it a viable option for eDNA studies in remote or resource-limited settings. In conclusion, addressing the limitations of eDNA techniques and implementing methods to improve species detection can lead to a more comprehensive assessment of biodiversity and a better understanding of the ecological impacts of anthropogenic changes. By maximizing DNA capture in the field, minimizing degradation during transport and storage, and ensuring successful isolation and amplification, eDNA techniques can provide valuable insights into biodiversity and ecosystem health.

## 5. Conclusions

The present study demonstrates the efficacy of the environmental DNA (eDNA) technique in assessing fish diversity in the challenging environment of canyon rivers. The findings from the Mabiehe River highlight the significant impact of dam constructions on local fish biodiversity, with dammed sections exhibiting reduced species richness and diversity compared to free-flowing river segments. The detection of a variety of species, including protected and exotic ones, underscores the comprehensive nature of eDNA as a monitoring tool. This approach not only complements traditional survey methods but also offers a more feasible and detailed understanding of fish communities in inaccessible and complex habitats. Our results advocate for the broader application of eDNA in biodiversity assessments and the need for conservation strategies that consider the ecological impacts of dam construction on riverine ecosystems.

## Figures and Tables

**Figure 1 animals-14-02433-f001:**
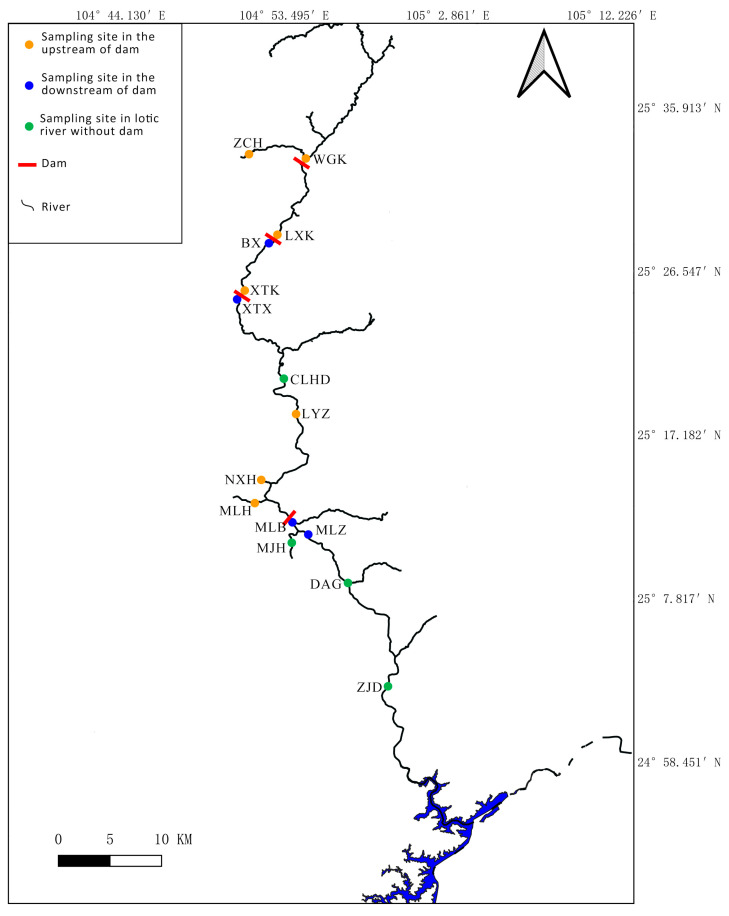
Map of sampling sites in the present study.

**Figure 2 animals-14-02433-f002:**
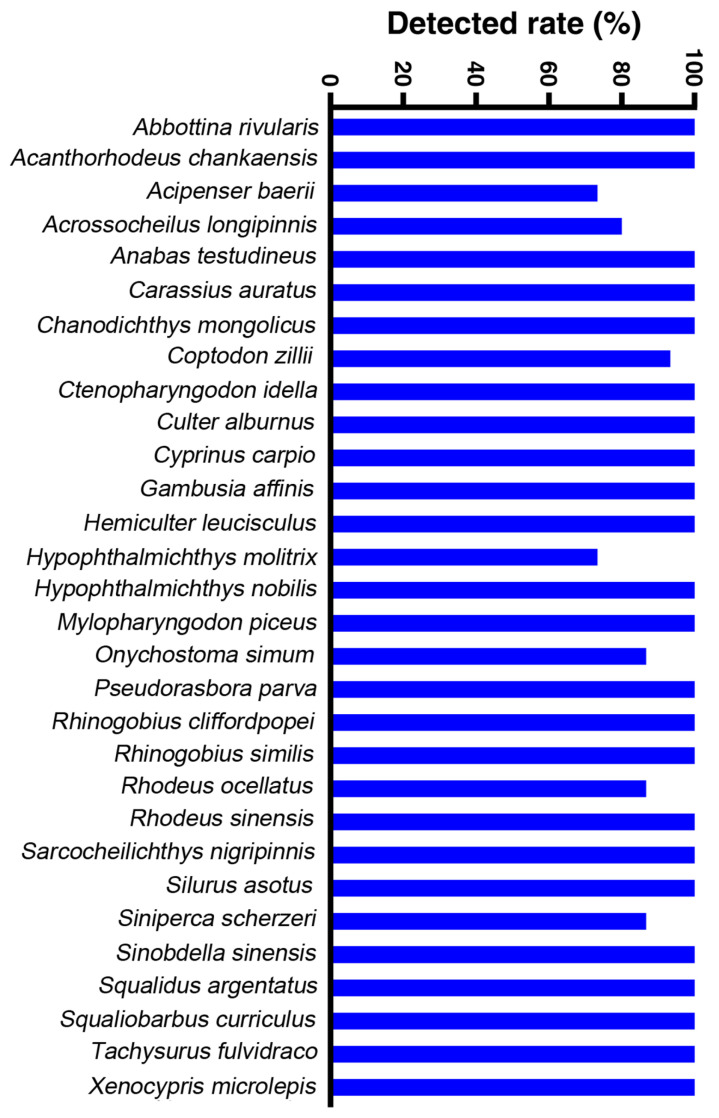
Top 30 detection rate of observed species.

**Figure 3 animals-14-02433-f003:**
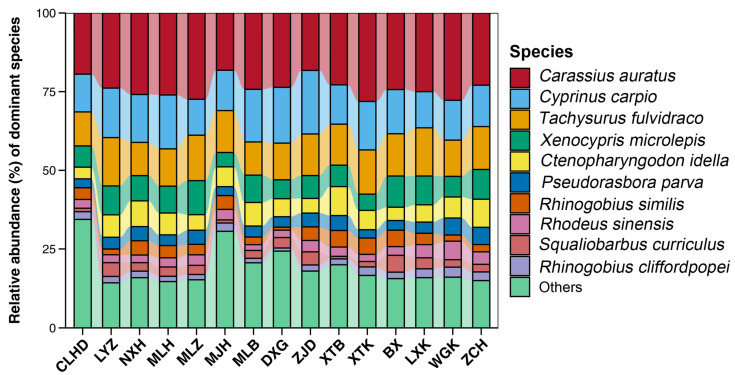
Species composition of dominant fish species based on the relative sequence abundance.

**Figure 4 animals-14-02433-f004:**
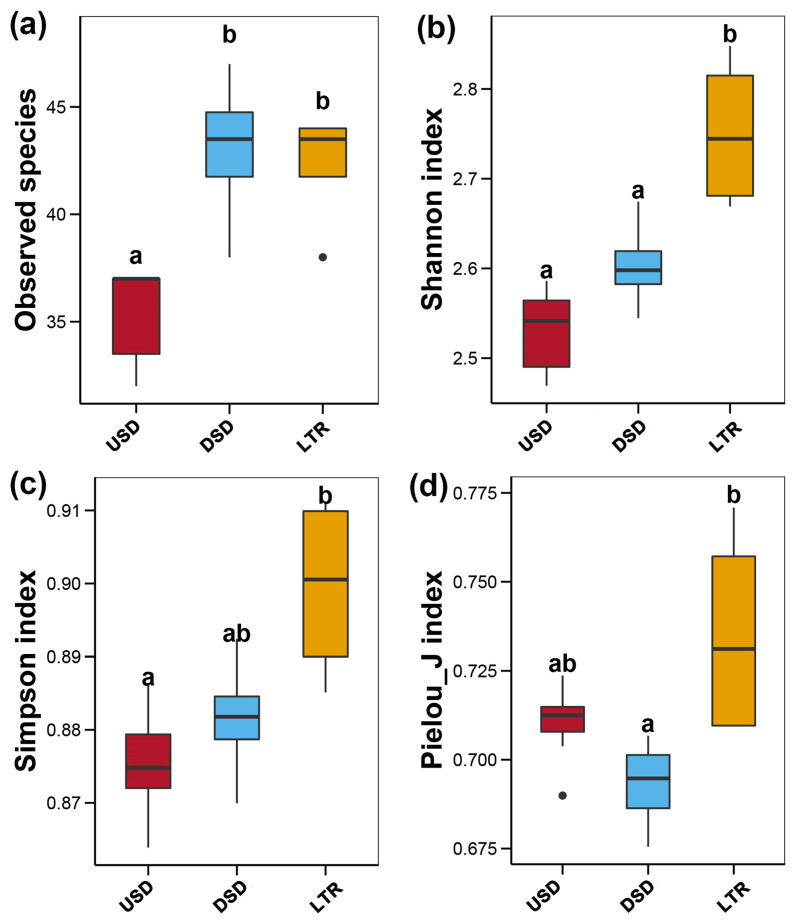
Species number (**a**) and alpha diversity indices (**b**–**d**) among upstream dams (USD), downstream dams (DSD), and lotic river without dams (LTR). Box plot representation: the horizontal line inside the box represents the median, and the lower and upper borders of the box represent the 25th and 75th percentiles, respectively. The upper and lower whiskers indicate the maximum and minimum range of the data excluding outliers (the points). Different lowercase letters indicate significant differences (*p* < 0.05) among different locations.

**Figure 5 animals-14-02433-f005:**
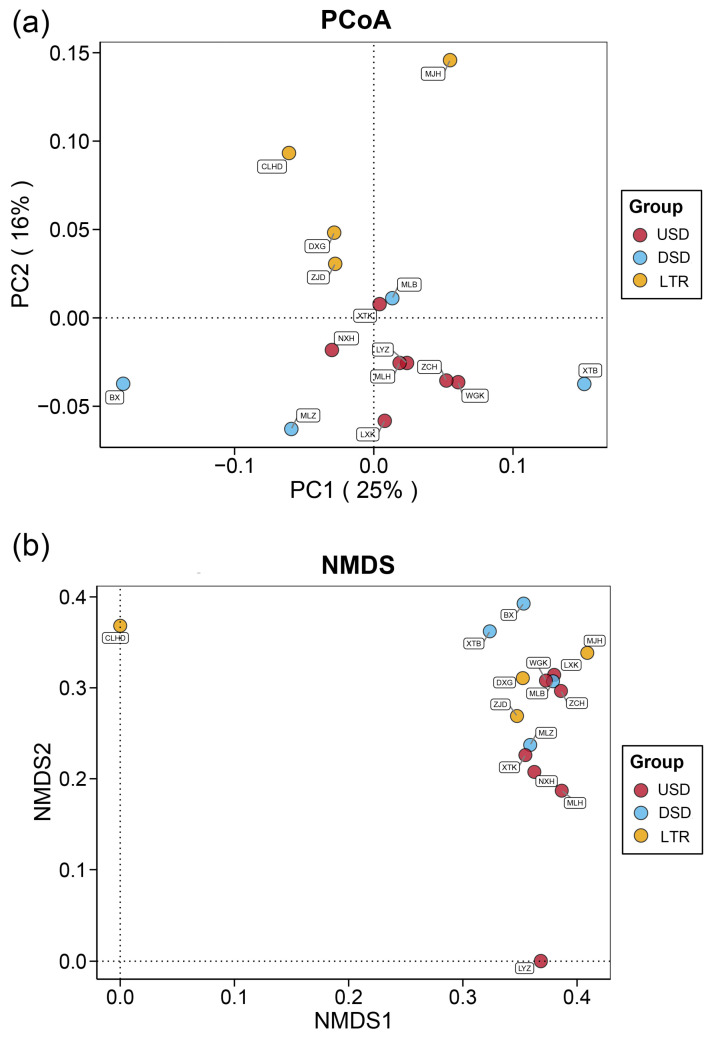
Principal co-ordinate analysis (**a**) and non-metric multidimensional scaling analysis (**b**) of each sample based on the Bray–Curtis distance.

**Table 1 animals-14-02433-t001:** Species lists and sequence numbers of observed fish based on eDNA at 15 sampling sites in the Mabiehe River.

Species	ZCH	WGK	LXK	BX	XTK	XTB	CLHD	LYZ	NXH	MLH	MLB	MJH	MLZ	DXG	ZJD
*Acipenser baerii*	373	0	0	211	536	972	225	482	400	0	1419	740	146	0	172
*Neosalanx taihuensis*	316	0	223	0	0	353	0	0	604	241	215	574	220	359	385
*Zacco platypus*	0	0	0	0	0	0	1	1	1	1	0	0	0	0	1
*Opsariichthys uncirostris*	222	255	0	0	315	290	169	321	0	244	0	0	0	213	193
*Ctenopharyngodon idella*	6584	5215	4004	2169	4060	8476	2918	5246	5228	5072	5506	5155	3017	3979	3496
*Mylopharyngodon piceus*	678	402	443	308	316	1479	878	592	631	759	232	418	394	1181	786
*Squaliobarbus curriculus*	1786	1849	2567	2755	1146	747	876	3224	1722	2160	1840	820	1782	2318	3126
*Abbottina rivularis*	172	99	162	143	100	78	102	85	73	191	216	254	103	85	328
*Pseudorasbora parva*	4040	4207	2587	1584	1770	4365	2215	2741	2882	2481	2579	2332	2742	2309	3291
*Hemibagrus macropterus*	0	0	212	0	0	0	238	0	0	0	0	0	0	0	0
*Hemibarbus labeo*	0	0	0	0	0	0	0	0	0	0	0	0	0	0	228
*Squalidus argentatus*	2	3	270	54	792	1180	175	617	852	582	260	815	551	685	821
*Saurogobio dabryi*	513	204	367	2	0	427	447	0	0	0	1	0	115	208	1
*Sarcocheilichthys nigripinnis*	689	1515	556	797	1031	711	1548	1112	1036	1086	2175	1856	290	2088	834
*Chanodichthys mongolicus*	244	11	20	183	810	562	430	517	346	592	625	219	553	407	644
*Culter alburnus*	86	1080	1099	376	751	202	805	263	161	870	745	673	363	593	56
*Hemiculter leucisculus*	29	299	258	447	187	53	497	35	24	29	481	499	39	40	762
*Xenocypris microlepis*	6973	5117	6695	5104	3426	6243	5290	6769	5126	6216	6398	3737	6607	4194	5447
*Hypophthalmichthys molitrix*	257	10	0	124	5	521	0	1	2	479	282	269	0	452	0
*Hypophthalmichthys nobilis*	783	217	227	8	728	520	413	7	668	241	41	38	370	21	6
*Cyprinus carpio*	9695	9886	8380	7229	10,209	11,485	9406	11,579	9746	12,477	12,308	10,508	6967	12,336	15,162
*Carassius auratus*	16,925	21,667	18,266	12,517	18,641	20,998	15,229	17,573	16,585	18,986	17,880	14,914	16,757	16,457	13,686
*Acrossocheilus kreyenbergii*	0	121	0	3	0	1	1	0	0	0	0	2	179	184	1
*Acrossocheilus longipinnis*	0	1	1	7	6	3	4	0	1	1	1	0	2	1	1
*Acrossocheilus parallens*	0	0	0	0	0	0	0	0	0	0	1	2905	0	0	0
*Acrossocheilus yunnanensis*	0	0	0	0	0	0	1216	0	0	0	0	0	0	0	0
*Spinibarbus sinensis*	1	716	0	1	0	0	242	308	0	0	244	2	166	486	0
*Onychostoma gerlachi*	0	0	1	5	2	1	9160	1116	1	1	1	798	709	1555	1536
*Percocypris pingi*	0	114	0	0	0	0	0	0	0	0	0	0	0	0	0
*Discogobio tetrabarbatus*	0	1	1	54	4	3975	949	0	0	0	1	0	0	2	2
*Semilabeo obscurus*	0	0	0	1	0	281	304	0	0	0	0	0	0	0	0
*Cirrhinus molitorella*	0	0	0	0	0	0	0	0	0	0	234	1	0	0	0
*Acanthorhodeus chankaensis*	1666	1413	1000	653	794	1759	2786	849	1086	1888	990	879	877	383	1222
*Acheilognathus tonkinensis*	0	0	1	0	0	0	0	0	0	0	0	0	0	176	0
*Rhodeus ocellatus*	274	53	1429	466	4	491	2	0	3	2	0	3	5	203	6
*Rhodeus sinensis*	2923	4583	3044	1417	1503	2706	2184	1861	1558	2128	1343	2746	2045	1640	2707
*Schizothorax lissolabiatus*	165	651	0	197	0	0	0	0	0	0	0	169	0	203	257
*Schistura fasciolata*	362	8	0	147	1	139	586	31	244	0	0	0	0	526	0
*Misgurnus anguillicaudatus*	1279	614	238	0	0	1	150	0	0	0	1	5753	552	1	438
*Paramisgurnus dabryanus*	0	3	0	149	1	259	255	0	0	0	238	0	2	1	428
*Sinibotia pulchra*	0	0	65	1	0	0	0	0	0	0	0	0	0	0	0
*Sinibotia robusta*	0	0	0	283	0	0	0	1	0	0	1	0	0	1	0
*Clarias gariepinus*	0	0	0	0	0	0	0	0	0	0	0	0	0	0	217
*Silurus asotus*	1004	2191	1208	1336	1583	625	1408	1590	1236	1855	1402	874	725	2129	1527
*Pseudobagrus crassilabris*	0	0	0	0	230	256	0	0	0	0	224	0	0	0	0
*Hemibarbus maculatus*	1	0	0	145	228	0	677	0	0	0	116	424	0	277	0
*Tachysurus fulvidraco*	10,037	8990	11,189	6900	9309	11,975	8444	11,300	6764	8626	7763	10,899	8825	8101	9917
*Glyptothorax fukiensis*	0	0	0	204	0	0	0	0	0	0	0	0	0	0	0
*Pareuchiloglanis longicauda*	0	0	0	246	0	0	595	0	0	0	0	0	0	0	0
*Ictalurus punctatus*	0	0	0	110	0	1	0	0	0	0	2	435	1	0	0
*Sinobdella sinensis*	468	1044	1454	315	1010	839	352	623	659	667	796	878	194	1130	570
*Gambusia affinis*	234	859	1038	592	1149	1215	1104	880	886	527	263	544	910	1423	1188
*Oryzias latipes*	0	0	232	1	299	1	144	261	550	0	237	0	313	2	196
*Rhinogobius cliffordpopei*	2028	2500	2055	1054	1798	1708	1915	1499	1334	1217	1052	2141	1021	691	1467
*Rhinogobius similis*	1701	1547	2654	2689	3428	4826	2905	1341	2906	2820	1808	3581	2027	660	3241
*Anabas testudineus*	62	79	135	63	84	72	53	99	164	57	55	70	74	73	221
*Channa maculata*	0	0	0	125	0	533	0	0	223	0	0	0	422	148	1
*Coptodon zillii*	708	551	145	11	19	294	1024	0	308	130	3292	4601	204	1330	241
*Lepomis cyanellus*	1	0	1	1	0	4	0	0	0	0	1	0	0	0	1
*Siniperca scherzeri*	434	0	801	241	4	276	1	682	0	223	377	396	810	356	210

**Table 2 animals-14-02433-t002:** Alpha diversity indices and species count of each sample.

Pop	Observed Species	Shannon	Simpson	Pielou_J
ZCH	37	2.5860	0.8864	0.7162
WGK	37	2.5413	0.8743	0.7038
LXK	37	2.5765	0.8810	0.7135
BX	47	2.6009	0.8820	0.6755
XTK	37	2.4912	0.8639	0.6899
XTB	44	2.6745	0.8924	0.7068
CLHD	44	2.8479	0.9095	0.7526
LYZ	33	2.4893	0.8748	0.7119
NXH	34	2.5520	0.8777	0.7237
MLH	32	2.4693	0.8698	0.7125
MLB	43	2.5951	0.8816	0.6900
MJH	38	2.8040	0.9112	0.7708
MLZ	38	2.5445	0.8699	0.6995
DXG	44	2.6849	0.8851	0.7095
ZJD	43	2.6690	0.8916	0.7096

## Data Availability

The data generated during and/or analyzed during the current study are available from the corresponding author on reasonable request.

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
