# Peer review of "Environment DNA Reveals Fish Diversity in a Canyon River within the Upper Pearl River Drainage"

_animals, 2024, doi:10.3390/ani14162433_

Round 1

Reviewer 1 Report

Comments and Suggestions for Authors

1.Line 49-50: the sentence "Canyon rivers often exhibit distinctive characteristics, such as steep gradients, swift water currents, abundant hydraulic resources, and intricate habitats" can be broken down or rephrased for clarity. Consider rephrasing "This attributes not only the focus of water resource management and hydropower development, but also serves as crucial habitats for indigenous fish species" for clarity and conciseness.

2. The phrase "render conventional fish survey methods arduous" could be rephrased to "make conventional fish survey methods challenging (Line 56-57)".

3.The sentence "Considering the challenging conditions of canyon rivers that render conventional fish surveys impractical" (Line 73-74) is somewhat repetitive of earlier points. It could be streamlined to avoid redundancy.

4.The transition between discussing the challenges of conventional methods and the introduction of eDNA techniques could be smoother. A brief explanation of how eDNA specifically addresses these challenges would strengthen the narrative.

5.The Materials and Methods section provides a comprehensive and detailed description of the study region, sampling procedures, sample preservation, DNA extraction, amplification, sequencing, and bioinformatics processing. This thoroughness ensures reproducibility of the study.

6.The statement "To prevent DNA degradation, the water samples were promptly stored in cold storage and filtered using a vacuum pump onto a 0.45 μm mixed cellulose filter membrane" could specify the temperature of the cold storage (Line 103-104). Also authors should Ensure consistency in the use of symbols and units, such as µL and ℃. 

Comments on the Quality of English Language

Minor editing of English language required

Author Response

  1. Line 49-50: the sentence "Canyon rivers often exhibit distinctive characteristics, such as steep gradients, swift water currents, abundant hydraulic resources, and intricate habitats" can be broken down or rephrased for clarity. Consider rephrasing "This attributes not only the focus of water resource management and hydropower development, but also serves as crucial habitats for indigenous fish species" for clarity and conciseness.

Response: done. Thank you very much for your valuable comment. We have improved them: “Canyon rivers often have unique features. These include steep gradients, fast-flowing currents, rich hydraulic resources, and complex habitats.” and “These characteristics make canyon rivers vital for both water resource management and hydropower development, as well as essential habitats for native fish species.”. Please see lines 48-51.

  1. The phrase "render conventional fish survey methods arduous" could be rephrased to "make conventional fish survey methods challenging (Line 56-57)".

Response: done. Thank you very much for your comment. Please see lines 57-58.

  1. The sentence "Considering the challenging conditions of canyon rivers that render conventional fish surveys impractical" (Line 73-74) is somewhat repetitive of earlier points. It could be streamlined to avoid redundancy.

Response: Thank you for your insightful feedback. We have carefully considered your suggestion regarding the sentence. To avoid redundancy and enhance clarity, we have streamlined the text in lines 53-71. Thank you for your valuable input.

  1. The transition between discussing the challenges of conventional methods and the introduction of eDNA techniques could be smoother. A brief explanation of how eDNA specifically addresses these challenges would strengthen the narrative.

Response: Thank you for your insightful comments on our manuscript. We appreciate your feedback and have revised the manuscript to address your concerns.

Regarding your comment on the transition between discussing the challenges of conventional methods and the introduction of eDNA techniques, we have improved the narrative to provide a smoother transition and a clearer explanation of how eDNA addresses these challenges. Below is the revised section of the manuscript (lines 57-71):

The formidable conditions in canyon rivers make conventional fish survey methods challenging. For instance, the steep and rugged terrain can make access to certain are-as difficult, while the fast-flowing currents pose a risk to both the equipment and the researchers. Additionally, the complex habitats with various submerged structures can impede the use of traditional netting and trapping techniques, often leading to incomplete or biased sampling. These conditions necessitate labor-intensive and invasive sampling methods that may not adequately capture the full diversity of fish species present.

To address these challenges, it is imperative to employ appropriate methodologies that can accurately assess fish diversity and the influence of dams on local fish populations in canyon rivers. Environmental DNA (eDNA) technique offers a promising alternative by effectively detecting the presence of the target organisms through the analysis of DNA released into the water. This noninvasive and sensitive method provides a significant advantage over conventional methods, particularly in difficult to access environments like canyon rivers.

We hope these revisions meet your expectations and improve the clarity and coherence of the manuscript. Thank you once again for your valuable feedback.

  1. The Materials and Methods section provides a comprehensive and detailed description of the study region, sampling procedures, sample preservation, DNA extraction, amplification, sequencing, and bioinformatics processing. This thoroughness ensures reproducibility of the study.

Response: Thank you for your feedback on our manuscript. We appreciate your positive assessment of the Materials and Methods section, particularly regarding its comprehensive description of the study region, sampling procedures, sample preservation, DNA extraction, amplification, sequencing, and bioinformatics processing. We have aimed for thoroughness in these aspects to ensure the reproducibility of our study. We will ensure that these methods are clearly detailed to facilitate replication by other researchers. Thank you once again for your valuable feedback.

  1. The statement "To prevent DNA degradation, the water samples were promptly stored in cold storage and filtered using a vacuum pump onto a 0.45 μm mixed cellulose filter membrane" could specify the temperature of the cold storage (Line 103-104). Also authors should Ensure consistency in the use of symbols and units, such as µL and ℃.

Response: Thank you for your valuable suggestion. We have specified the temperature of the cold storage in line 118 and reviewed the manuscript thoroughly to ensure consistency in the use of symbols and units.

Reviewer 2 Report

Comments and Suggestions for Authors

The core of the paper is an investigation into the fish diversity of the Mabiehe River, a canyon river in the upper reaches of the Pearl River in Guizhou Province, China, using eDNA techniques. The study identified 60 freshwater fish species, including nationally and provincially protected species, as well as invasive species. It highlighted that free-flowing sections of the river maintained higher species richness and diversity compared to dammed sections. Additionally, the research validated the efficacy of eDNA as a robust tool for complementing traditional field surveys, particularly suitable for biodiversity monitoring in inaccessible canyon rivers. The paper on fish diversity presents several strong points but also areas that could be improved for greater scientific rigor and clarity. Here are detailed critiques and suggestions for revision:

-1- The paper could benefit significantly from a more comprehensive discussion on the limitations of the eDNA technique. While eDNA offers a non-invasive and potentially expansive overview of biodiversity, its inability to detect all species present in the ecosystem is a critical limitation. This may be due to various factors including primer specificity, DNA degradation, or the dilution effect in large water bodies. To enhance the manuscript, it would be prudent to discuss potential methods to improve species detection. This includes using multiple sets of primers targeting different taxa or genetic markers, and optimizing sampling protocols to capture a broader range of DNA present in the environment. Such enhancements could provide a more comprehensive assessment of biodiversity and the ecological impacts of anthropogenic changes like dam construction.

2--The paper lacks comparative analysis with previous biodiversity assessments using traditional methods before the implementation of eDNA. Incorporating a temporal comparison would underscore the efficacy of eDNA in detecting shifts or stabilities in species diversity over time. It would also help validate eDNA as a reliable method against more established traditional survey techniques. The authors should consider integrating historical data on fish diversity from previous studies and discuss any observed discrepancies or confirmations between the datasets.

3--The decision to collect water samples exclusively from a depth of 0.5 meters may not adequately represent the full vertical biodiversity profile of the river, especially in a canyon river system where species stratification can occur due to varying environmental conditions at different depths. The authors should address how this sampling depth was chosen and discuss the potential biases it might introduce. Expanding the sampling depth range could be suggested to ensure a more representative capture of the river’s biodiversity.

4--The paper currently does not thoroughly discuss the rationale behind selecting the 12S rRNA region of the mitochondrial genome as the target gene for eDNA analysis. This section could be expanded to explain why this particular marker was chosen, its benefits, and any limitations it may have compared to other potential genetic markers. This could include a discussion on the marker’s resolution for species identification, its susceptibility to environmental degradation, and its ability to detect both broad and specific taxonomic groups.

5-The boxplot in Figure 4 lacks clear descriptions of what the data points represent. It is crucial for the authors to specify what each symbol (e.g., dots, crosses) in the boxplot denotes, whether they represent outliers, means, medians, or specific percentile values. Providing a legend or a clearer annotation within the figure can enhance reader understanding and ensure the graphical data presentation is transparent and informative.

Author Response

The core of the paper is an investigation into the fish diversity of the Mabiehe River, a canyon river in the upper reaches of the Pearl River in Guizhou Province, China, using eDNA techniques. The study identified 60 freshwater fish species, including nationally and provincially protected species, as well as invasive species. It highlighted that free-flowing sections of the river maintained higher species richness and diversity compared to dammed sections. Additionally, the research validated the efficacy of eDNA as a robust tool for complementing traditional field surveys, particularly suitable for biodiversity monitoring in inaccessible canyon rivers. The paper on fish diversity presents several strong points but also areas that could be improved for greater scientific rigor and clarity. Here are detailed critiques and suggestions for revision:

-1- The paper could benefit significantly from a more comprehensive discussion on the limitations of the eDNA technique. While eDNA offers a non-invasive and potentially expansive overview of biodiversity, its inability to detect all species present in the ecosystem is a critical limitation. This may be due to various factors including primer specificity, DNA degradation, or the dilution effect in large water bodies. To enhance the manuscript, it would be prudent to discuss potential methods to improve species detection. This includes using multiple sets of primers targeting different taxa or genetic markers, and optimizing sampling protocols to capture a broader range of DNA present in the environment. Such enhancements could provide a more comprehensive assessment of biodiversity and the ecological impacts of anthropogenic changes like dam construction.

Response: Thank you for your feedback on our manuscript. We appreciate your insightful comments and have made revisions accordingly to enhance the quality of our work. We have expanded the discussion section to address the inability of eDNA to detect all species present in the ecosystem. We have elaborated on factors such as primer specificity, DNA degradation, and the dilution effect in large water bodies; and potential methods to improve species detection. These information is now discussed in detail in the revised manuscript lines 360-430.

-2- The paper lacks comparative analysis with previous biodiversity assessments using traditional methods before the implementation of eDNA. Incorporating a temporal comparison would underscore the efficacy of eDNA in detecting shifts or stabilities in species diversity over time. It would also help validate eDNA as a reliable method against more established traditional survey techniques. The authors should consider integrating historical data on fish diversity from previous studies and discuss any observed discrepancies or confirmations between the datasets.

Response: Thank you for your insightful comments and the opportunity to revise our manuscript. We appreciate your suggestions for improving the manuscript.

The Mabiehe River, characterized by steep gradients, fast-flowing currents, rich hydraulic resources, and complex habitats, presents formidable conditions that make conventional fish survey methods challenging. The river hosts ten dams, which further complicate traditional fish surveys. To date, only one relevant study has been conducted in this river by Wang et al. (2023), which covers the region from CLHD to MLZ (as illustrated in Figure 1 of our manuscript). In comparison, our eDNA analysis detected a significantly higher number of species. Specifically, the traditional fishing methods used by Wang et al. (2023) failed to detect 42 fish species that were identified using eDNA in our study. Only the following species were detected in both studies: Abbottina rivularis, Pseudorasbora parva, Hemibarbus labeo, Hemiculter leucisculus, Cyprinus carpio, Carassius auratus, Acrossocheilus yunnanensis, Onychostoma gerlachi, Percocypris pingi, Discogobio tetrabarbatus, Semilabeo obscurus, Schizothorax lissolabiatus, Schistura fasciolata, Misgurnus anguillicaudatus, Paramisgurnus dabryanus, Silurus asotus, Pareuchiloglanis longicauda and Rhinogobius similis. This significant discrepancy highlights the effectiveness of eDNA in monitoring fish biodiversity, especially in challenging environments like canyon rivers  where traditional methods may fall short. However, our eDNA analysis did not detect four fish species (Rectoris longifinus, Sinocrosscheilus tridentis, Sinocyclocheilus angustiporus, and Pterocryptis anomala) that were identified by Wang et al. (2023). These species typically have smaller biomass and may be underrepresented in eDNA samples due to their specific habitat preferences and lower overall abundance.

This comparative approach validates the efficacy of eDNA in complementing traditional field surveys by overcoming the limitations posed by the river's steep gradients, fast currents, and complex habitats. The higher detection rate of species through eDNA underscores its utility in providing a more comprehensive understanding of the fish community in such challenging environments. 

We have added the information in lines 255-274. We hope that these revisions meet your expectations and enhance the overall quality of our manuscript. Thank you once again for your valuable feedback.

-3-The decision to collect water samples exclusively from a depth of 0.5 meters may not adequately represent the full vertical biodiversity profile of the river, especially in a canyon river system where species stratification can occur due to varying environmental conditions at different depths. The authors should address how this sampling depth was chosen and discuss the potential biases it might introduce. Expanding the sampling depth range could be suggested to ensure a more representative capture of the river’s biodiversity.

Response: Thank you for your valuable feedback regarding the sampling depth in our study. We appreciate your concern about the potential biases introduced by collecting water samples exclusively from a depth of 0.5 meters, especially in a canyon river system where species stratification may occur.

We would like to clarify that in our study, we did not limit our sampling to a single depth. Instead, we employed a comprehensive approach by collecting water samples from three different depths: the surface, 0.5 meters, and the lower layer. These samples were then mixed to create a composite sample that more accurately represents the vertical biodiversity profile of the river. This method was chosen to ensure that we capture a wide range of species that might inhabit different strata within the water column.

By mixing water samples from various depths, we aimed to mitigate the potential biases associated with sampling from a single depth. This approach helps in providing a more representative capture of the river’s biodiversity, considering the potential variations in species distribution due to environmental conditions at different depths.

We hope this clarification addresses your concerns. We have included this information in the revised manuscript (Please see lines 106-110) to ensure that the methodology is clearly communicated to readers.

Thank you once again for your insightful comments.

-4-The paper currently does not thoroughly discuss the rationale behind selecting the 12S rRNA region of the mitochondrial genome as the target gene for eDNA analysis. This section could be expanded to explain why this particular marker was chosen, its benefits, and any limitations it may have compared to other potential genetic markers. This could include a discussion on the marker’s resolution for species identification, its susceptibility to environmental degradation, and its ability to detect both broad and specific taxonomic groups.

Response: Thank you for your insightful comments regarding the selection of the 12S rRNA region of the mitochondrial genome for eDNA analysis. We appreciate the opportunity to clarify and expand on this point.

In the revised manuscript, we have elaborated on the rationale behind choosing the 12S rRNA gene as the target for eDNA analysis. Specifically, we highlighted that past research has demonstrated that primers targeting the 12S rRNA gene generally outperform those targeting other genes, such as 16S rRNA or COI, in terms of amplified fish diversity (Kumar et al., 2022; Shan Zhang et al., 2020). We selected the fish universal primers Tele02-F (5′-AAACTCGTGCCAGCCACC-3′) and Tele02-R (5′-GGGTATCTAATCCCAGTTTG-3′) targeting the 12S rRNA region of the mitochondrial genome for PCR amplification for this study. These primers, an improved version of the “MiFish-U” primer, demonstrate enhanced taxa resolution and increased specificity (Taberlet et al., 2018). This improvement is critical for accurate and comprehensive biodiversity assessments (Lines 127-133).

We have discussed limitations of primer selection in species identification and suggested to employ multiple primers to detect both broad and specific taxonomic groups (Lines 378-404).

References:

Kumar, G., Reaume, A. M., Farrell, E., & Gaither, M. R. (2022). Comparing eDNA metabarcoding primers for assessing fish communities in a biodiverse estuary. PLoS One, 17(6), e0266720.

Taberlet, P., Bonin, A., Zinger, L., & Coissac, E. (2018). Environmental DNA: For biodiversity research and monitoring: Oxford University Press.

Zhang, S., Zhao, J., & Yao, M. (2020). A comprehensive and comparative evaluation of primers for metabarcoding eDNA from fish. Methods in Ecology and Evolution, 11(12), 1609-1625.

-5-The boxplot in Figure 4 lacks clear descriptions of what the data points represent. It is crucial for the authors to specify what each symbol (e.g., dots, crosses) in the boxplot denotes, whether they represent outliers, means, medians, or specific percentile values. Providing a legend or a clearer annotation within the figure can enhance reader understanding and ensure the graphical data presentation is transparent and informative.

Response: Thank you for your feedback on our manuscript. We have provided clearer annotation in the figure as follows: Box-plot representation: the horizontal line inside the box represents the median, and the lower and upper borders of the box represent the 25th and 75th percentiles, respectively. The upper and lower whiskers indicate the maximum and minimum range of the data excluding outliers (the points).

Reviewer 3 Report

Comments and Suggestions for Authors

The introduction effectively sets the stage for the study by providing a comprehensive background on the challenges of surveying fish diversity in canyon rivers and the potential impacts of dam construction. It clearly justifies the use of eDNA techniques and outlines the specific context of the Mabiehe River. However, addressing certain weaknesses—particularly in terms of literature gaps, detailed challenges, objective clarity, and reference integration—would further strengthen the introduction. Specific comments are provided below:

1.       Line 54-57: While the introduction mentions the limited empirical evidence regarding the impacts of dams on fish diversity, it could benefit from a more detailed discussion of specific gaps in the current literature. Highlighting these gaps would further emphasize the study's contribution to this field.

2.       Line 64-68: The challenges of conventional fish surveys in canyon rivers are mentioned but could be elaborated further. Providing specific examples of these challenges would give readers a clearer understanding of why eDNA is a preferable method in such environments.

The Materials and Methods section is detailed and provides a solid framework for understanding the study’s methodological approach. To further strengthen this section, it is important to address identified weaknesses, particularly in the areas of DNA quantification, PCR optimization, sequencing depth, data analysis pipeline, and normalization procedure. Overall, the methodologies employed are appropriate and thorough, supporting the study’s objectives and findings. However, in lines 102-109, while the use of a 0.45 μm mixed cellulose filter membrane is mentioned, more details on the filtration process would be beneficial. For instance, specifying the volume of water filtered per membrane and the duration of the filtration process would provide a more comprehensive understanding.

The results section is well-structured and presents a clear and detailed account of fish composition, and alpha and beta diversity analyses in the study area. The inclusion of both protected and alien species findings, along with the comparison across different areas, adds significant value to the study. Further discussion on ecological implications and methodological details would strengthen the overall impact of the results. Specific comments are provided below:

1.       The use of figures and tables to illustrate species distribution and abundance is effective. Figures 2 and 3, in particular, provide a clear visual representation of the presence and relative abundance of species across sampling sites, aiding in the comprehension of the results.

2.       Line 201-206: Additional context or discussion on the ecological implications of the observed diversity patterns would enhance the reader's understanding. For example, exploring why LTR exhibits higher diversity and evenness, or the potential factors driving the distinct community compositions observed in PCoA and NMDS, would provide deeper insights.

The discussion sections are well-structured and provide valuable insights into the fish composition, community dynamics, and the performance of the eDNA technique in the Mabiehe River. Further elaboration on the ecological implications of the detected fish composition, including the potential impacts of invasive species on native biodiversity and ecosystem services, would enhance the reader's understanding. Furthermore, including a brief discussion on the potential long-term ecological consequences of reduced fish diversity in dammed sections, such as effects on ecosystem services and local fisheries, would enhance the relevance and applicability of the findings. Additionally, comparing fish diversity between eDNA and traditional methods or historical records would deepen reader’s understanding eDNA’s application in this field.

Other comments:

1. Line 50: These attributes......

2. Line 51: Removed the comma after "water resource management and hydropower development" for clarity.

3. Line 64: Changed "only one pertinent study" to "only one relevant study" for better word choice.

4. Line 92: Clarified "sampling sites were conducted" to "sampling sites were established."

5. Line 162: sampling locations.

6. Line 165-166: demonstrated the highest species abundance, accounting for 58.33%.

7. Line 167-168: nationally protected fish species.

Author Response

The introduction effectively sets the stage for the study by providing a comprehensive background on the challenges of surveying fish diversity in canyon rivers and the potential impacts of dam construction. It clearly justifies the use of eDNA techniques and outlines the specific context of the Mabiehe River. However, addressing certain weaknesses—particularly in terms of literature gaps, detailed challenges, objective clarity, and reference integration—would further strengthen the introduction. Specific comments are provided below:

  1. Line 54-57: While the introduction mentions the limited empirical evidence regarding the impacts of dams on fish diversity, it could benefit from a more detailed discussion of specific gaps in the current literature. Highlighting these gaps would further emphasize the study's contribution to this field.

Response: Thank you for your insightful comments on our manuscript. We appreciate the opportunity to address your suggestions and have provided a more comprehensive and detailed introduction: There is insufficient data on the long-term impacts of dams on fish diversity. Studies provide snapshots of biodiversity at single points in time, without examining how fish communities change before and after dam construction, or how they might fluctuate over extended periods. Please see lines 54-57.

  1. Line 64-68: The challenges of conventional fish surveys in canyon rivers are mentioned but could be elaborated further. Providing specific examples of these challenges would give readers a clearer understanding of why eDNA is a preferable method in such environments.

Response: Thank you for your valuable feedback and suggestions to improve our manuscript. We appreciate your insights and have addressed your comments accordingly (lines 57-64): The formidable conditions in canyon rivers present several challenges for conventional fish survey methods. For instance, the steep and rugged terrain can make access to certain areas difficult, while the fast-flowing currents pose a risk to both the equipment and the researchers. Additionally, the complex habitats with various submerged structures can impede the use of traditional netting and trapping techniques, often leading to incomplete or biased sampling. These conditions necessitate labor-intensive and invasive sampling methods that may not adequately capture the full diversity of fish species present. As such, environmental DNA (eDNA) emerges as a preferable method in such environments due to its non-invasive nature and ability to provide comprehensive biodiversity assessments without the need for direct physical capture of organisms.

The Materials and Methods section is detailed and provides a solid framework for understanding the study’s methodological approach. To further strengthen this section, it is important to address identified weaknesses, particularly in the areas of DNA quantification, PCR optimization, sequencing depth, data analysis pipeline, and normalization procedure. Overall, the methodologies employed are appropriate and thorough, supporting the study’s objectives and findings. However, in lines 102-109, while the use of a 0.45 μm mixed cellulose filter membrane is mentioned, more details on the filtration process would be beneficial. For instance, specifying the volume of water filtered per membrane and the duration of the filtration process would provide a more comprehensive understanding.

Response: Thank you for your constructive feedback and for appreciating the methodologies employed in our study. We have provided additional details on the filtration process to address your comments on lines 112-119. Thank you once again for your valuable feedback.

The results section is well-structured and presents a clear and detailed account of fish composition, and alpha and beta diversity analyses in the study area. The inclusion of both protected and alien species findings, along with the comparison across different areas, adds significant value to the study. Further discussion on ecological implications and methodological details would strengthen the overall impact of the results. Specific comments are provided below:

  1. The use of figures and tables to illustrate species distribution and abundance is effective. Figures 2 and 3, in particular, provide a clear visual representation of the presence and relative abundance of species across sampling sites, aiding in the comprehension of the results.

Response: Thank you for your positive feedback on the use of figures and tables in our manuscript. We appreciate your acknowledgment and are glad that these visual aids have contributed to the clarity of our findings. Thank you once again for your valuable comments and support.

  1. Line 201-206: Additional context or discussion on the ecological implications of the observed diversity patterns would enhance the reader's understanding. For example, exploring why LTR exhibits higher diversity and evenness, or the potential factors driving the distinct community compositions observed in PCoA and NMDS, would provide deeper insights.

Response: Thank you for your insightful comments. We have expanded our discussion to include these aspects, specifically addressing why the lotic river (LTR) section exhibits higher diversity and evenness in Discussion (lines 316-325): The observed higher diversity and evenness in the LTR section can be attributed to its status as a flowing river section that is less affected by dam-related disturbances. The original ecological conditions in the habitats are largely maintained, allowing for a more stable and diverse community of species. In contrast, sections of the river influenced by damming often experience altered flow regimes, habitat fragmentation, and changes in water quality, all of which can reduce species diversity and disrupt community composition. The distinct community compositions observed in the PCoA and NMDS analyses further reflect these environmental gradients and the varying degrees of anthropogenic impact across different sections of the river.

We hope this additional discussion provides deeper insights into the ecological implications of our findings.

The discussion sections are well-structured and provide valuable insights into the fish composition, community dynamics, and the performance of the eDNA technique in the Mabiehe River. Further elaboration on the ecological implications of the detected fish composition, including the potential impacts of invasive species on native biodiversity and ecosystem services, would enhance the reader's understanding.

Response: thanks a lot for your suggestion. We have expanded our discussion according to your suggestions as below:

Given that C. zillii and G. affinis are two widespread invasive species in China [41-48], they can outcompete native fish for resources, alter habitat structures, and disrupt ex-isting food webs. This can lead to a decline in native species populations and a loss of biodiversity. These changes can compromise ecosystem services such as water purifi-cation, nutrient cycling, and recreational fisheries, ultimately affecting the ecological integrity and sustainability of the river ecosystem. Increased efforts in developing targeted management strategies to protect native biodiversity and maintain ecosystem functions should be made to minimize their invasion areas as much as possible (Lines 293-300).

Furthermore, including a brief discussion on the potential long-term ecological consequences of reduced fish diversity in dammed sections, such as effects on ecosystem services and local fisheries, would enhance the relevance and applicability of the findings.

Response: Thank you for your constructive feedback on the manuscript (lines 335-352).

Dams are obstructing rivers worldwide, impairing habitat and migration opportunities for many freshwater fish species. They remove turbulent river sections and create tranquil water bodies, thus affecting flow and temperature regimes, sediment transport, and species communities. The shift from lotic to lentic environments after dam construction often favors generalist species over specialists, altering assemblages of taxonomic groups and putting endemic species at particular risk of extinction, leading to biotic homogenization [53,54]. Shifts in temperature regimes, including downstream decreases in temperature due to hypolimnic releases from reservoirs, impair conditions favorable to native species but may favor exotics or habitat generalists [55].

The long-term ecological consequences of reduced fish diversity in dammed sections are significant. These changes can lead to impaired ecosystem services, such as nutrient cycling, water purification, and sediment transport, essential for maintaining ecosystem health. Furthermore, the reduction in biodiversity can negatively impact local fisheries, which rely on a diverse and stable fish population for sustainable yields. This can lead to economic losses for communities dependent on fishing for their livelihoods and diminish food security. Understanding these long-term consequences is crucial for developing effective conservation strategies and mitigating the adverse impacts of dam construction on river ecosystems [56].

Additionally, comparing fish diversity between eDNA and traditional methods or historical records would deepen reader’s understanding eDNA’s application in this field.

Response: Thank you for your feedback on the discussion sections of our manuscript. We have added the information in the discussion, please see lines 255-274.

Other comments:

  1. Line 50: These attributes......

Response: done. We have rephrased to “These characteristics ”. Please see line 49.

  1. Line 51: Removed the comma after "water resource management and hydropower development" for clarity.

Response: thank you very much for your comments. We have rephrased the sentence for more clarity: “These characteristics make canyon rivers vital for both water resource management and hydropower development, as well as essential habitats for native fish species”. Please see lines 49-51.

  1. Line 64: Changed "only one pertinent study" to "only one relevant study" for better word choice.

Response: done. Please see line 77.

  1. Line 92: Clarified "sampling sites were conducted" to "sampling sites were established."

Response: done. Please see line 100.

  1. Line 162: sampling locations.

Response: done. Please see line 175.

  1. Line 165-166: demonstrated the highest species abundance, accounting for 58.33%.

Response: done. Please see line 178.

  1. Line 167-168: nationally protected fish species.

Response: done. Please see line 180.